# Detection of *Helicobacter pylori* from Extracted Teeth of a Patient with Idiopathic Thrombocytopenic Purpura

**DOI:** 10.3390/microorganisms10112285

**Published:** 2022-11-17

**Authors:** Masakazu Hamada, Ryota Nomura, Saaya Matayoshi, Yuko Ogaya, Hiroyasu Kameyama, Narikazu Uzawa, Kazuhiko Nakano

**Affiliations:** 1Department of Oral and Maxillofacial Surgery II, Osaka University Graduate School of Dentistry, Osaka 565-0871, Japan; 2Department of Pediatric Dentistry, Osaka University Graduate School of Dentistry, Osaka 565-0871, Japan; 3Department of Pediatric Dentistry, Hiroshima University Graduate School of Biomedical and Health Sciences, Hiroshima 734-8553, Japan

**Keywords:** immune thrombocytopenic purpura, *H. pylori*, oral surgery, molecular biological analysis, extracted tooth

## Abstract

Immune thrombocytopenic purpura (ITP) is an autoimmune disease characterized by isolated cryptogenic thrombocytopenia due to a transient or persistent reduction in platelet count. Many patients with ITP have shown improved platelet count after *Helicobacter pylori* eradication therapy. However, there have been no studies regarding *H. pylori* in the oral cavity of patients with ITP. Here, we describe a patient with ITP whose oral samples exhibited *H. pylori*. A 64-year-old woman with ITP came to our hospital with chief complaints that required oral surgery, including tooth extraction and cystectomy. Bacterial DNA from *H. pylori* was confirmed on the extracted tooth, but was not detected in the saliva taken at the time. Bacterial DNA from *H. pylori* was detected on the suture around the extraction socket, which was removed at 10 days post-operation. However, *H. pylori* DNA was not detected in other oral samples at 10 or 30 days post-operation. A urea breath test was carried out in the gastrointestinal clinic at 60 days post-operation, which revealed no presence of *H. pylori* in the gastrointestinal tract. These results suggest that teeth with severe bacterial infections may be a potential reservoir of *H. pylori* for patients with ITP.

## 1. Introduction

Immune thrombocytopenic purpura (also known as idiopathic thrombocytopenic purpura; ITP) is an autoimmune disease characterized by isolated thrombocytopenia with fewer than 100,000/µL platelets due to a transient or persistent reduction in platelet count [1]. Although ITP is presumably caused by the autoimmune destruction of platelets in the spleen, the detailed mechanism underlying this phenomenon remains unknown. When the platelet count falls below 50,000/µL due to ITP, bleeding may be evident [2]. Furthermore, when the platelet count drops to less than 10,000/µL, severe bleeding symptoms such as oral bleeding, hematuria, and intracranial hemorrhage may occur [3,4,5,6,7]. When these symptoms develop, corticosteroids, high-dose gamma-globulin therapy, platelet infusion, and splenectomy should be considered [3,8].

*Helicobacter pylori* is a Gram-negative microaerophilic human gastric pathogen with strong urease activity and a helical structure [9]. *H. pylori* infection, which causes chronic gastritis, peptic ulcer disease and gastric cancer [9], can be diagnosed via the histological detection of bacteria in gastric biopsies, urea breath test and stool antigen test [10].

Associations between *H. pylori* and extragastric diseases, such as neurological diseases, respiratory diseases, and hematologic diseases, have been previously reported [11,12]. In 1998, it was reported that the number of platelets increased in patients with ITP after the eradication of *H. pylori* [13]. In addition, studies focusing on the relationship between ITP and *H. pylori* infection have been reported in recent years [14,15], and the eradication of *H. pylori* is regarded as an effective treatment for ITP [14,15]. Examination of *H. pylori* in gastric infections in patients with ITP is typically performed by non-invasive tests such as the urea breath test and stool antigen test; invasive tests (e.g., endoscopic biopsy) are not recommended due to the difficulty involved in maintaining hemostasis [16].

*H. pylori* was previously classified as a *Campylobacter* species, which are known to be major periodontopathic bacteria [9]. Therefore, the oral cavity is a possible route of *H. pylori* infection, and a potential reservoir for the bacteria [17]. There are approximately 700 bacterial species in the oral cavity, and it is not possible to specifically detect *H. pylori* from the oral cavity by the same methods used to detect *H. pylori* from gastric tissue. To detect *H. pylori* in the oral cavity, molecular microbiological analyses such as polymerase chain reaction (PCR) have been widely used [18].

The detection rate of *H. pylori* by PCR from the oral cavity has been reported to be 0–100% [19], and it is not easy to identify the actual infection rate. To overcome the difficulty of detecting *H. pylori* from oral specimens, we designed a novel primer set based on the whole-genome information of about 50 *H. pylori* strains in our database [20,21]. Among the entire genome of *H. pylori*, 16S rRNA, *vacA*, *cagA*, *glmM* (*ureC*), and *ureA* have been used frequently for *H. pylori* detection by PCR [22,23,24,25,26]. We searched for homology between all *H. pylori* in each of these genes. As a result, only *ureA* had six consecutive sequences of 20 or more bases conserved in the approximately 50 *H. pylori* strains [20], and we selected some of these regions as primer sets for *H. pylori* detection.

Although molecular biological analysis is used as a tool for diagnosis of *H. pylori* infection in the oral cavity, there have been no published studies regarding *H. pylori* detection in oral samples taken from patients with ITP. In the present report, we describe the detection of *H. pylori* in the oral cavity of a patient with ITP who underwent oral surgery comprising tooth extraction and cystectomy.

## 2. Materials and Methods

### 2.1. Ethics Statement

This study was conducted in full adherence to the Declaration of Helsinki. This study protocol was approved by the Ethics Committee of Osaka University Graduate School of Dentistry (approval: H27-E17-1). Written consent to participate in this study was obtained from the subject patient. The patient first came to our clinic on 19 October 2017, underwent surgery on 13 December 2017, and the consultation at our clinic was completed on 12 January 2018.

### 2.2. DNA Extraction

Bacterial DNA was extracted using previously described methods [20]. Briefly, oral specimens were resuspended in 250 μL of 10 mM Tris-HCl (pH 8.0) containing 100 mM NaCl and 1 mM EDTA. The cells were collected using centrifugation and lysed in 600 µL of Cell Lysis Solution (Qiagen, Düsseldorf, Germany) and incubated at 80 °C for 5 min, followed by the addition of 3 μL of RNase A (10 mg/mL; Qiagen) and incubation at 37 °C for 30 min. Protein Precipitation Solution (Qiagen) was added, vigorously vortexed for 20 s, and then centrifuged at 10,000× *g* for 3 min. The supernatant was combined with 600 μL of isopropanol (Wako Pure Chemical Industries, Tokyo, Japan) and centrifuged. The precipitate was then resuspended in 70% ethanol (Wako Pure Chemical Industries), centrifuged, combined with 100 μL of TE buffer (10 mM Tris-HCl, 1 mM EDTA [pH 8.0]), and used as bacterial DNA for the PCR method.

### 2.3. PCR Detection of Bacteria

We previously searched the whole-genome sequences of about 50 *H. pylori* strains in a database to construct a PCR method to detect *H. pylori* from oral specimens with high sensitivity and specificity [20]. We found that more than 20 nucleotides of sequences were consecutively conserved in all *H. pylori* strains at multiple locations in the *ureA* gene. We designed primer sets for *H. pylori* detection using some of these sequences in the *ureA* gene (Table 1).

PCR with either of these forward and reverse primer combinations detected 10^2^ – 10^3^ colony forming units (CFU) of *H. pylori* [21]. To further improve the detection sensitivity, we developed a nested PCR method. In the nested PCR, the first-step PCR for *H. pylori* detection was performed using primers *ureA*-aF and *ureA*-bR, and the second-step PCR (nested PCR) using primers *ureA*-bF and *ureA*-aR (Figure 1).

For the first-step PCR, 2 μL of bacterial DNA extracted from oral samples was used in a 20 μL reaction. For the second-step PCR, 1 μL of the first-step PCR amplification product was used as a template for the reaction in a total volume of 20 μL. Both the first and second steps of nested PCR were performed as described previously, using TaKaRa Ex *Taq* polymerase (Takara Bio. Inc., Otsu, Japan) [21]. During each nested PCR reaction, sterile distilled water was used in place of bacterial DNA as a negative control to ensure that no false positive reactions occurred. PCR amplification was performed with the following cycling parameters: initial denaturation at 95 °C for 4 min, followed by 30 amplification cycles (95 °C for 30 s, 55 °C for 30 s, and 72 °C for 30 s), and a final extension at 72 °C for 7 min. The amplicon size of the first-step PCR product was 488 bp and the amplicon size of the second-step PCR product was 383 bp [21].

## 3. Case Report

A 64-year-old woman with ITP was presented to the Department of Oral and Maxillofacial Surgery at Osaka University Dental Hospital for tooth extraction and cystectomy due to a radicular cyst. She had been diagnosed with ITP in the Department of Hematology three years prior. The patient’s platelet count was stable at approximately 30–50 × 10^3^/μL (Figure 2); thus, the hematologist considered her to be at low risk of hemorrhagic diathesis, and she received a routine dental check-up.

In an intraoral examination, temporary sealing with cement was observed on the right mandibular first molar (Figure 3). Although intraoral examination showed no obvious gingival swelling around the roots of the right mandibular molar region, panoramic radiography showed a cystic lesion around the root apices of the right mandibular first molar (Figure 4A–C). Therefore, we selected tooth extraction and cystectomy as treatment. Although the patient’s preoperative platelet count was 36 × 10^3^/μL (Table 2), which is below the normal range, we performed the tooth extraction and cystectomy under intravenous sedation without platelet transfusion on the basis of the hematologist’s opinion that the patient was at low risk of hemorrhagic diathesis (Figure 4D). No abnormal bleeding was observed in the socket, and the operative site was packed with Surgicel^®^ (Ethicon Inc., New Brunswick, NJ, USA) and suture; a splint was then placed for hemostasis. The patient received 1 g of cefazolin intravenously during tooth extraction and 100 mg of cefditoren pivoxil, three times a day after each meal, for three days after tooth extraction. The patient did not use an antimicrobial mouthwash such as chlorhexidine after tooth extraction.

To analyze whether *H. pylori* was present in the patient’s oral cavity, samples of the extracted tooth were taken intra-operatively with the patient’s consent (Figure 5). In addition, 1 mL of unstimulated saliva was collected in sterile plastic tubes intra-operatively and post-operatively (after 10 and 30 days). A suture and splint were included in the analysis; these were removed at 10 days post-operation. Finally, dental plaque samples were taken from four teeth (upper left second premolar, lower left second premolar, lower right second premolar, and lower right second molar) during a follow-up visit at 30 days post-operation. The patient was not given any restrictions such as brushing their teeth or eating before the collection of oral samples.

The first-step PCR did not detect *H. pylori* DNA in the oral specimens. the second-step PCR detected *H. pylori* DNA on the extracted teeth, but not in the saliva taken intra-operatively (Figure 6). In addition, the second-step PCR detected *H. pylori* on the suture used at the extracted tooth site, which was removed from the site 10 days post-operation, whereas *H. pylori* was not detected in saliva or on the splint taken on the same day. Moreover, *H. pylori* was not detected in dental plaque samples from four teeth or in saliva samples taken 30 days post-operation. The bands obtained by the nested PCR method were sequenced using previously described methods [21,27], and were confirmed to correspond to *H. pylori* genes.

Based on these results, we referred the patient to a gastroenterologist to examine the presence of *H. pylori* in the gastrointestinal tract. On day 60 post-operation, the gastroenterologist performed a urea breath test using urea labeled with ^13^C, which returned a negative result for *H. pylori*.

During the follow-up at our hospital, one-month post-operative panoramic radiography showed no problems (Figure 7). A family dentist performed subsequent prosthetic treatment and follow-up. More than four years have passed since the tooth extraction, and there have been no visits to our clinic due to post-operative problems.

## 4. Discussion

An association between ITP and *H. pylori* infection was first reported in 1998 [13]. Within the past 20 years, a large number of studies have demonstrated that *H. pylori* is an etiological factor in the onset of ITP [14,15]. Platelet count was reported to be significantly increased by 3.5–4.2-fold after *H. pylori* eradication therapy in patients with ITP [14,15]. However, all prior studies were focused on the relationship between ITP and *H. pylori* infection in gastric tissues. Therefore, we examined whether *H. pylori* was present in the oral cavity of a patient with ITP.

We investigated the distribution of *H. pylori* in oral samples from the patient, such as saliva and the extracted tooth. Detection of *H. pylori* in these oral samples was performed by nested PCR using *H. pylori* detection primer sets targeting the *ureA* gene, as in our previous study [20]. Nested PCR showed *H. pylori* DNA on the extracted tooth, but not in saliva taken intra-operatively. In addition, saliva samples taken at 10 and 30 days post-operation were negative for *H. pylori*, as were dental plaque samples taken from four different teeth at 30 days post-operation. However, *H. pylori* DNA was detected on a suture used in the extraction socket, which was removed at 10 days post-operation. These results indicate that *H. pylori* was localized to the extracted tooth but was not present in saliva or on most other oral surfaces.

The diagnosis of *H. pylori* infection includes bacterial culture, urea breath test, urease test, histology, and serology results [28]. However, *H. pylori* detection methods are generally designed to detect *H. pylori* from gastric tissues, and cannot detect *H. pylori* in the oral cavity, where about 700 bacterial species exist [20]. Therefore, molecular biological analyses such as PCR, real-time PCR, and nested PCR methods are widely used to detect *H. pylori* in the oral cavity [20]. Even using these molecular biological methods, it is difficult to detect *H. pylori* accurately and sensitively from the oral cavity [20,21]. We previously examined various detection methods that were less likely to produce false-positive results [20,21]. As a result, we confirmed that the nested PCR method specifically detected only *H. pylori*, and did not detect closely related species including *H. felis* or *H. pullorum*, demonstrating that the incidence of false positives was extremely low. The nested PCR method also has high sensitivity for the detection of *H. pylori*; as few as 1–10 CFU were detected, even in the presence of other oral bacteria and in oral tissues from oral specimens [21]. Therefore, more than 1–10 CFU of *H. pylori* were present in the extracted tooth and sutures of the ITP patient. The nested PCR method used in the present study detected *H. pylori* in oral specimens including the saliva, dental plaque, extracted teeth, and inflamed dental pulp [29,30,31].

Recently, it was reported that a unique oral microbiome forms in subjects under special systemic conditions including disease or congenital anomalies [32]. Furthermore, the oral conditions of individuals in many developing countries in Asia are worse than those in Japan [33,34]. The unique oral environment greatly influences *H. pylori* colonization in the oral cavity. For example, the association between periodontal disease and oral *H. pylori* infection has attracted attention. *H. pylori* colonize in periodontal pockets formed by periodontal disease [35], and periodontal treatment reduces *H. pylori* reinfection into the stomach [36]. *H. pylori* and *Porphyromonas gingivalis*, a major periodontopathic bacteria, are often detected together in the oral cavity [37]. *H. pylori* is also more likely to be detected in the presence of a specific periodontopathic bacterial group called the red complex (*Porphyromonas gingivalis*, *Treponema denticola*, and *Tannerella forsythia*) and less likely to be detected in the presence of the orange complex (*Prevotella intermedia*, *Prevotella nigrescens*, and *Campylobacter rectus*) or green complex (*Capnocytophaga ochracea*, *Capnocytophaga sputigena*, and *Eikenella corrodens*) [37]. In the present case report, *H. pylori* colonized the cystic lesion around the root apices of an ITP patient, and the specific environment or oral bacterial species present in the lesion may be involved in the colonization of *H. pylori*. These results suggest that the prevention of dental disease is essential in controlling *H. pylori* colonization.

Table 3 summarizes all of the studies that have used the same nested PCR method for *H. pylori* detection as used in this study. Among those papers, the oral specimen with the highest *H. pylori* detection rate was the inflamed dental pulp, with a positivity rate of 38.9% [21]. In contrast, in the other papers, the *H. pylori* detection rate from dental pulp ranged from 2.6% to 12.0% [31,37], and that from extracted teeth ranged from 9.2% to 17.9% [29,37]. The lower detection rate of *H. pylori* in these pulps and extracted teeth than in the inflamed dental pulp may be due to the inclusion of teeth with periodontal diseases that did not develop pulpal infection. In our case report, a panoramic X-ray finding showed no alveolar bone resorption, and the cause of the radicular cyst may be reinfection of the pulp that had undergone root canal treatment. Therefore, preventing the development of dental caries-induced inflamed pulp is crucial, as it could serve as a reservoir for *H. pylori* in ITP.

In the nested PCR used in this study, *H. pylori* DNA was not detected in all oral specimens in the first-step PCR. In the second-step PCR, *H. pylori* DNA was detected in the extracted tooth and in the sutures used at the tooth extraction site on the 10th post-operative day. In a study using inflamed pulp from 131 subjects, the detection rate of *H. pylori* in first-step PCR was 3.1% [21]. In contrast, the detection rate of *H. pylori* in second-step PCR using identical specimens was 38.9% [21]—significantly higher than that in first-step PCR. These results suggest that oral lesions mainly contain enough *H. pylori* to be detected by second-step PCR but not by first-step PCR.

The oral cavity is a possible reservoir for *H. pylori*, and thus plays a crucial role in both *H. pylori* transmission and gastric infection [38]. In the present case, the patient completed a urea breath test after tooth extraction, which revealed that no *H. pylori* was present in the gastrointestinal tract. *H. pylori* may not have been detected in the gastrointestinal tract because it was present only in a limited portion of the extracted tooth, and was not present in the gastrointestinal tract before the tooth extraction. Conversely, *H. pylori* may have been present in both the extracted tooth and the gastrointestinal tract; thus, *H. pylori* was eliminated by extraction of the tooth that constituted a reservoir for bacterial colonization. If possible, a urea breath test should have been performed before tooth extraction to demonstrate the presence of *H. pylori* in the gastric tissue. In future studies, a larger number of patients with ITP should be analyzed, focusing on the presence of *H. pylori* in both the oral cavity and the gastrointestinal tract, to confirm whether eradication of *H. pylori* from the oral cavity leads to its eradication from the gastrointestinal tract.

We prescribed 1 g of cefazolin intravenously during tooth extraction and 100 mg of cefditoren pivoxil, three times a day after each meal, for three days after tooth extraction. The main purpose of antibiotic administration after tooth extraction is to prevent infection. However, β-lactam antibiotics such as cefazolin and cefditoren pivoxil are also used to eradicate *H. pylori* [30], and they may be effective at eradicating *H. pylori* remaining in the tooth extraction sockets.

This study had some limitations. First, our data have only highlighted a simple casual relationship between oral *H. pylori* and ITP. To clarify the many unknown factors involved, a longer follow-up of a large number of patients is required to determine the association between *H. pylori* in the oral cavity and a clinical improvement in ITP. We should monitor changes in patient condition after the removal of *H. pylori,* including improved clinical findings of ITP and reinfection of *H. pylori*. In addition, the relationship between *H. pylori* in the oral cavity and that in the gastric tissues needs to be clarified. Second, since the subject had no subjective symptoms in the digestive system and her platelets were decreased, the subject requested a urea breath test, which is a non-invasive and simple test. However, if possible, a definitive diagnosis should be performed by culture tests for *H. pylori* using tissue samples from gastroscopy. In addition, antibody testing and histopathologic analysis of *H. pylori* should be performed. Third, since our treatment was completed after tooth extraction, we only had information on platelet count immediately after tooth extraction. Previous reports have shown that platelet count significantly increases in ITP patients 6 months after eradication treatment [14,15]. Therefore, data on platelet count should be collected for at least 6 months, even after removal of *H. pylori* in the oral cavity.

## 5. Conclusions

*H. pylori* DNA was detected in the oral cavity, localized around the extracted tooth of a patient with ITP. Although the causal relationship between the oral detection of *H. pylori* and ITP is still unknown, teeth with severe bacterial infections, such as radicular cysts, are potential reservoirs of *H. pylori*. Based on these findings, dental treatment or oral care that leads to the eradication or reduction of *H. pylori* from the oral cavity is recommend in patients with diseases such as ITP.

## Figures and Tables

**Figure 1 microorganisms-10-02285-f001:**
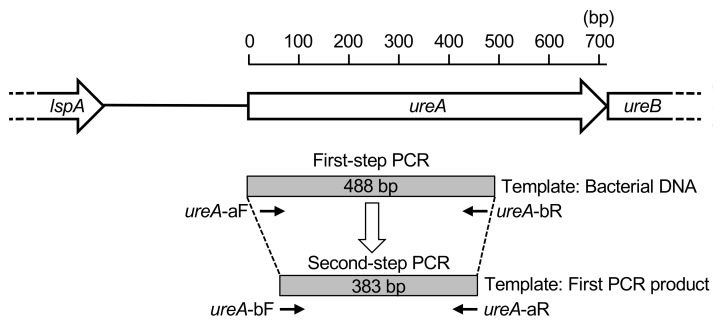
Schematic diagram of positions of the designed primers for *Helicobacter pylori* detection.

**Figure 2 microorganisms-10-02285-f002:**
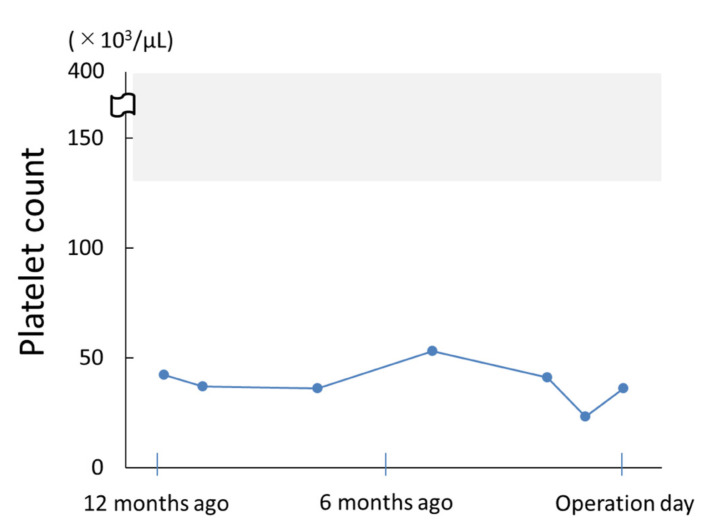
Platelet counts in the year leading up to the surgery. The gray color indicates the standard value for healthy subjects.

**Figure 3 microorganisms-10-02285-f003:**
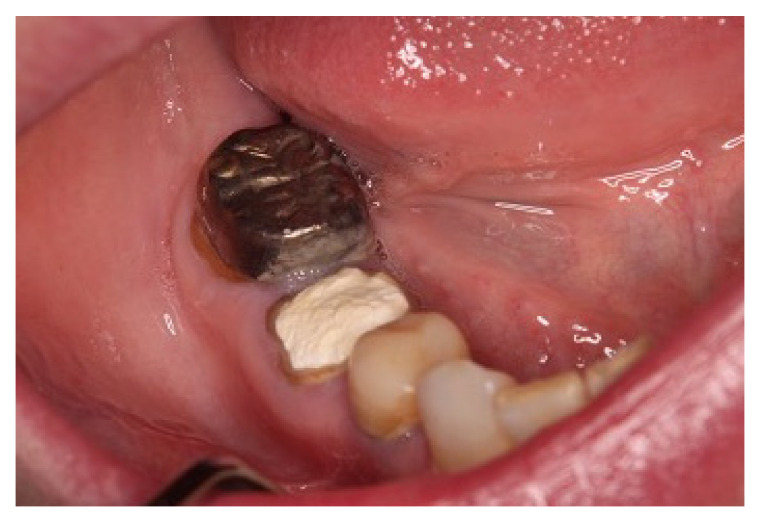
Intraoral photographs taken at the first visit to our hospital. The first molar was temporarily sealed with cement.

**Figure 4 microorganisms-10-02285-f004:**
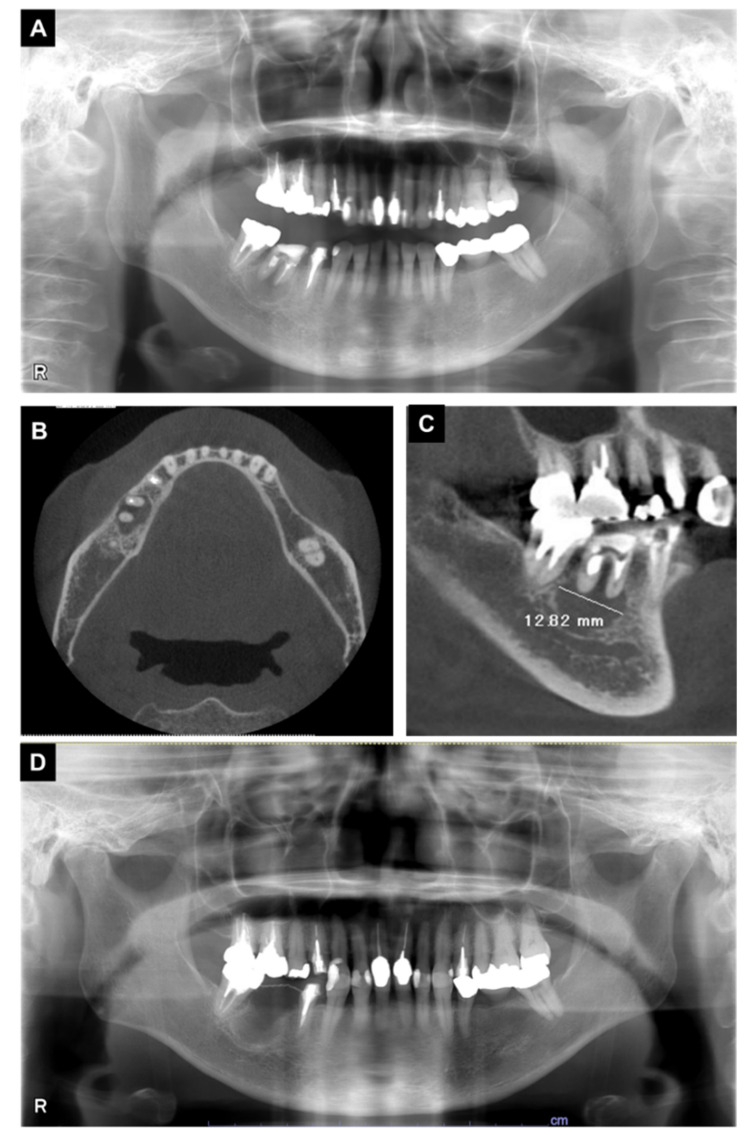
X-ray examination findings during the initial visit. (**A**) Panoramic radiography; (**B**,**C**) computed tomography images. Radiolucent findings were observed at the root apex of the first molar. (**D**) Panoramic radiography images after surgery.

**Figure 5 microorganisms-10-02285-f005:**
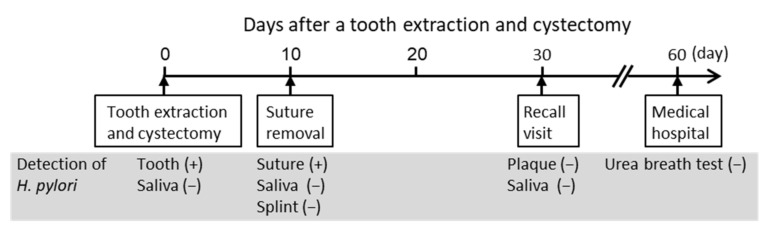
Timeline of the present case after tooth extraction and cystectomy.

**Figure 6 microorganisms-10-02285-f006:**
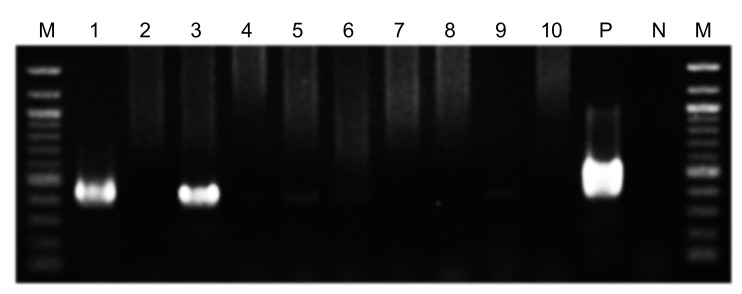
Detection of *Helicobacter pylori* in oral samples. Polymerase chain reaction results showing the second-step PCR assay using *ureA*-bF and *ureA*-aR primer sets for detection of *H. pylori* in oral samples. Lanes: 1, extracted tooth; 2, saliva taken immediately pre-operation; 3, suture taken 10 days post-operation; 4, saliva taken 10 days post-operation; 5, splint taken 10 days post-operation; 6, dental plaque (upper left second premolar) taken 30 days post-operation; 7, dental plaque (lower left second premolar) taken 30 days post-operation; 8, dental plaque (lower right second premolar) taken 30 days post-operation; 9, dental plaque (lower right second molar) taken 30 days post-operation; 10, saliva taken 30 days post-operation; P (positive control), *H. pylori* ATCC 43504; N (negative control), sterile water; M, 100-bp DNA ladder.

**Figure 7 microorganisms-10-02285-f007:**
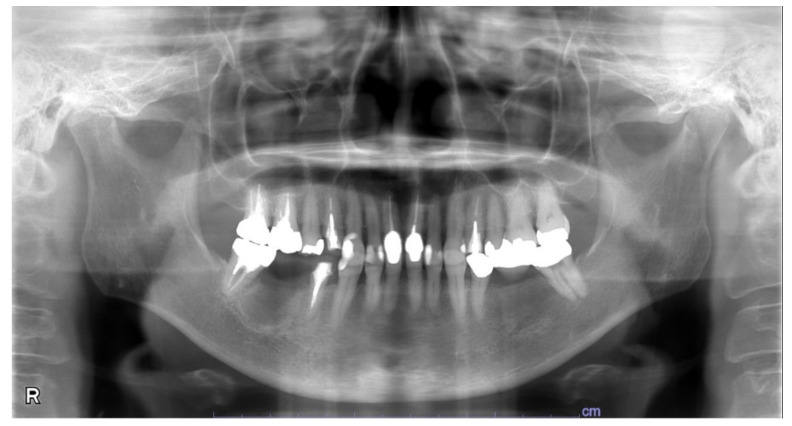
A panoramic radiography image taken during the one-month follow-up period after surgery.

**Table 1 microorganisms-10-02285-t001:** *H. pylori* detection primers.

Primer	Sequence (5′–3′)	Size (bp)	References
First step PCR			
*ureA*-aF	ATG AAA CTC ACC CCA AAA GA	488	[21]
*ureA*-bR	CCG AAA GTT TTT TCT CTG TCA AAG TCT A		
Second step PCR			
*ureA*-bF	AAA CGC AAA GAA AAA GGC ATT AA	383	[21]
*ureA*-aR	TTC ACT TCA AAG AAA TGG AAG TGT GA		

**Table 2 microorganisms-10-02285-t002:** Results of the preoperative blood analysis.

Item	Abbreviation	Value	Standard Value
Red blood cell count	RBC	4.5 × 10^6^/μL	3.8–4.8 × 10^6^/μL
White blood cell count	WBC	4.1 × 10^3^/μL	4.0–9.0 × 10^3^/μL
Platelet count	Plt	36 × 10^3^/μL	130–400 × 10^3^/μL
Hemoglobin	Hb	13.1 g/dL	12.0–16.0 g/dL
Activated partial thromboplastin time	APTT	28.6 s	22.0–39.0 s
Prothrombin time	PT (s)	12.7 s	10.0–13.0 s
Prothrombin time activity	PT (%)	89.10%	80.0–120.0%
Prothrombin time-international normalized ratio	PT-INR	1.1	1.0 ± 0.15

**Table 3 microorganisms-10-02285-t003:** Previous studies of *H. pylori* detected in the oral cavity, by the same nested PCR method as used in this study.

Study No.	Oral Specimen	Number of Subjects (Age)	History of Gastrointestinal Disease	Detection Rates of *H. pylori*	References
1	Inflamed dental pulp	n = 131 (1–19 years)	Unknown	38.90%	[21]
2	Dental plaque (Extracted tooth)	n = 87	Digestive diseases	9.20%	[29]
	Saliva	(20–83 years;	(n = 16)	14.90%	
		mean 45.6 years)			
3	Dental pulp	n = 192	Urine antibody test positive (n = 25)	12.00%	[31]
	Dental plaque	(mean 58.6 years)		1.00%	
	Saliva			0%	
4	Dental plaque	n = 1 (29 years)	Stomachache and Urine antibody test positive	100%	[30]
	Saliva			100%	
5	Dental pulp	n = 39 (16–70 years;	Previous *H. pylori* infection in gastric tissue (n = 4)	2.60%	[37]
	Dental plaque (Extracted tooth)	mean 35.3 years)		17.90%	
	Saliva			5.10%	

## Data Availability

Not applicable.

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
