# Peer review of "Detection of Helicobacter pylori from Extracted Teeth of a Patient with Idiopathic Thrombocytopenic Purpura"

_microorganisms, 2022, doi:10.3390/microorganisms10112285_

Round 1

Reviewer 1 Report

This is an interesting study but has some serious flaws which unfortunately make it hard for me to recommend publication. The presented data cannot possibly support the conclusion.

Whereas I have no problems with the overall presentation of the results or in any way dispute the findings, I find the fact that this refers to a single case reduces its relevance. I think that the authors themselves show the general problem in table 3 the and in their discussion. Small sample numbers and low incidence of H. pylori make it difficult to ascertain exactly why the patient sample was positive. Was it, as the authors suggest, that the tooth acted as a reservoir for H. pylori or was it contamination? The trouble with PCR based methods is that their sensitivity can give rise to false positives for a number of reasons. What is the source of the infection since the patient is otherwise H. pylori negative? There is also the question of whether the removal of the tooth and maybe other associated infections had a positive effect on the underlying condition. Even if there is can this be associated with the removal of the H. pylori?  In the other published work it appears that there was an ongoing H. pylori infection that was eradicated. In this case there is no evidence other than the PCR result, of an ongoing infection. Was there for instance any sign of H. pylori antibodies? What other organisms were associated with the cyst.

In short I think that the data is too thin and this submission is premature. I think that the authors have to survey a larger sample. to ascertain whether H. pylori can be detected in the oral cavity in the absence of an ongoing gastric infection and if so whether this can be associated with pathology.

Reviewer 2 Report

Congratulations for this research, which was well designed and performed!

I hope you will continue studies, which could bring major benefits to medical practice.

Reviewer 3 Report

The manuscript consists of total 11 pages, including 5 figures, 3 tables and the list of total 37 literature references. The manuscript is a case study with a significant background theory presentation, considering the option that the presented case of idipathic trombocytopenic purpura possibly may have been associated by the coexisting Helicobacter pylori persistent presence in the oral cavity structures, including the removed tooth - with the limitation that a longer follow-up is needed to confirm the expected clinical improvement and draw conclusions about the actual association of both conditions in the reported case. As such, the article fits into the scope of the works published in the Journal and can be considered as one of high clinical education importance. The manuscript is written in correct, communicative and narrative English and as such it presents as a educationally useful and pleasurable read. The title of the manuscript is adequate to its contents.

The Abstract section mirrors both the structure and contents of the main text of the article adequately.
The Introduction section presents enough background of the investigated scientific problem.
The actual case report / results / section is detailed enough, supported richly aand adequately with graphic material.
The text misses a separate Material and methods section - instead, the respective information is mixed into the case report. It would be advisable that the Authors create a Material and methods section with a header and move all the respective fragments of the article into it for the sake of the discipline and clarity of the text.
The Discussion section adequately places the presented case report into the broader existing knowledge context.
The Conclusions relate to the discussed case report and are careful enough; in my opinion the Authors shall repeat in this section the information that a longer follow-up of the patient is needed to draw any more binding conclusions about the clinical improvement and confirmation of association between the previous presence of Helicobacter pylori in the mouth tissues, including the extracted tooth, and the idipathic trombocytopenic purpura.
The literature references are numerous enough and relevant to the topic of the manuscript; some of them quite old but in case of a quite rare condition can be still considered as recent enough.

Reviewer 4 Report

The authors attribute the detection of H. pylori in the patient’s oral cavity to the compromised tooth (right mandibular first molar) since, after the tooth extraction/cystectomy the microorganism was not detected anymore. However, there is no comment about other possible factors related to this event such as the use of antimicrobials in the postoperative period. Did the patient use any systemic antibiotic or antimicrobial mouthwash such as chlorhexidine after the tooth extraction? A comment about that should be raised in the discussion session. Also, in the case description the authors should make clear whether this kind of medication was used or not.

Reviewer 5 Report

Dear Authors 

Thanks for the opportunity of reviewing your paper 

Despite the good quality of the presentation it is not clear to me the causal relation between the detection of Helicobacter pylori from Extracted Teeth and the suture and the overall clinical condition of the presented patient.

The patient was negative to Helicobacter detection in plaque and saliva and no data about bacterial concentration in the extracted tooth are given. A simple casual relation seems to exit from the given data with a scarce clinical significance

Reviewer 6 Report

I congratulate the authors for their highly valuable case study. I'm looking forward to studes with more patients confirming this interesting results.

Round 2

Reviewer 3 Report

Since my previous remarks have been followed by the Authors I do not have any more comments on the manuscript to add.

Reviewer 4 Report

No comments.

Reviewer 5 Report

Dear Authors 

as you appointed is just a casual and not causal relation

Round 3

Reviewer 5 Report

Dear Authors

the highlighted issues have been addressed
